# Exploring the Impact of Passive Ankle Exoskeletons on Lower-Limb Neuromechanics during Walking on Sloped Surfaces: Implications for Device Design

James L. Williamson [1,*], Glen A. Lichtwark [2] and Taylor J. M. Dick [1]

1   School of Biomedical Sciences, University of Queensland, Brisbane, QLD 4072, Australia; t.dick@uq.edu.au
2   School of Exercise & Nutrition Sciences, Queensland University of Technology, Brisbane, QLD 4000, Australia; glen.lichtwark@qut.edu.au
*   Correspondence: james.williamson@uq.net.au

**Abstract:** Humans and animals navigate complex and variable terrain in day-to-day life. Wearable assistive exoskeletons interact with biological tissues to augment movement. Yet, our understanding of how these devices impact the biomechanics of movement beyond steady-state environments remains limited. We investigated how passive ankle exoskeletons influence mechanical energetics and neuromuscular control of the lower-limb during level, incline, and decline walking. We collected kinematic and kinetic measures to determine ankle, knee, and hip mechanics and surface electromyography to characterize muscle activation of lower-limb muscles while participants walked on level, incline, and decline surfaces ($0°$, $+5°$, and $−5°$) with exoskeletons of varying stiffnesses ($0$–$280$ Nm rad$^{-1}$). Our results demonstrate that walking on incline surfaces with ankle exoskeletons was associated with increased negative work and power at the knee and increased positive work and power at the hip. These alterations in joint energetics may be linked to an additional requirement to load the springy exoskeleton in incline conditions. Decline walking with ankle exoskeletons had no influence on knee or hip energetics, likely owing to disrupted exoskeleton clutch actuation. To effectively offload the musculoskeletal system during walking on sloped surfaces, alterations to passive ankle exoskeleton clutch design are necessary.

**Keywords:** biomechanics; locomotion; plantarflexors; spring-loaded; wearable assistive devices





## 1. Introduction

Given the critical role of the ankle joint for human locomotor efficiency [1] and stability [2], numerous wearable devices have targeted assistance at this distal lower-limb joint [3–5]. Despite size and strength differences in the muscles that cross the ankle in comparison to the knee and hip joints, the ankle is a locus for assistance because it provides the majority of total lower-limb power during steady-state walking [6,7]. Several ankle exoskeletons have demonstrated their ability to reduce the whole-body energetic demands of level walking [3,5,8–13]. A category of these devices, passive ankle exoskeletons, reduce metabolic demands by storing and returning elastic strain energy via spring-clutch components—a mechanical analogue of the catapult mechanism within the ankle plantarflexors and Achilles tendon. Previous studies that have used passive ankle exoskeletons to augment walking have been conducted on level, flat surfaces [5,8,14]. However, humans navigate complex and variable environments in everyday life. Currently, our understanding of how passive ankle exoskeletons impact lower-limb mechanics and neuromuscular control in situations where energy must be generated or dissipated remains limited. One such example of these conditions is locomotion on incline and decline surfaces.

During level walking, spring-loaded ankle exoskeletons have been shown to influence locomotor performance across several scales—at the level of the whole-body [5,8,12], joint [5,8], and individual muscle [15]. The ability for passive ankle exoskeletons to reduce

whole-body metabolic cost, by way of optimizing the timing and magnitude of assistance, has historically been a key focus of investigation [5,8,12]. At the joint level, increasing exoskeleton stiffness alters ankle kinematics, with higher stiffnesses associated with more plantarflexed postures [5,8]. In addition, the biological contribution of the ankle joint to the total ankle moment (biological + exoskeleton) has been shown to decrease with increasing stiffness [5,8]—demonstrating the ability of devices to offload the muscle–tendon units that cross the ankle joint. It has been suggested that the likely driver of whole-body metabolic reductions during assisted walking is the reduction in active muscle volume of the plantarflexor muscles [16]. With exoskeleton assistance, the activation of soleus (SOL)—the largest of the plantarflexors—decreases during mid-stance [5,8,14]. However, decreased SOL activity is met with a concomitant increase in tibialis anterior (TA) activation during the swing phase [5,8]. This increased TA activation, when coupled with a 'trade-off' between improved SOL economy in mid-stance and reduced SOL economy in late-stance [15], may help to explain why increasing device stiffness does not systematically lead to reductions in metabolic cost. In fact, a 'bowl-shaped' relationship between device stiffness and the net metabolic rate has been established across various walking speeds [5,8], whereby an effective device stiffness range of 50–80 Nm rad$^{-1}$ results in the greatest energetic reductions. However, the relationship between device stiffness and muscle activation across a broader range of locomotor conditions remains unexplored.

The mechanical demands placed on the lower-limb while moving in uphill and downhill environments varies depending on the steepness of the slope and may limit the effectiveness of springy ankle exoskeletons that store and return energy [17]. During incline and decline locomotion, net positive or net negative work must be performed, respectively, on the bodies' center of mass (COM), owing to the change in the height of the COM between strides. During incline walking, this results in a shift in the mechanical contributions of the lower-limb joints whereby, positive power production moves from the ankle to the hip with increasing grade [17,18]. Simultaneously, negative power at the ankle reduces with increasing grade [19]. The shift in the site of positive power generation from the ankle to the hip suggests that providing exoskeleton assistance at the ankle during incline locomotion may not be as consequential as assistance during level walking. Furthermore, a reduction in negative power at the ankle during incline walking fundamentally limits the capacity of ankle-based devices to load their elastic elements and store energy. As such, users may require alternative sources of biological energy to load a springy exoskeleton. During decline walking, the knee emerges as the dominant site of energy dissipation, acting to absorb more energy than during level walking [17,18]. These grade-dependent redistributions of lower-limb mechanical behavior disrupt the energetic motivations for devices that assist the ankle. Exploring how humans adapt lower-limb behavior with varying exoskeleton stiffness across tasks where net COM work must be generated or absorbed (i.e., incline and decline walking) will provide fundamental insights to inspire the future design of assistive devices capable of performing in more complex, real-world environments.

In this study, we determined how passive ankle exoskeletons of varying stiffnesses influence the mechanical energetics and neuromuscular control of the lower-limb during level, incline, and decline walking. To do this, participants walked at 1.25 m s$^{-1}$ on level, incline, and decline surfaces (0°, +5°, −5°) with passive ankle exoskeletons of varying stiffnesses (0–280 Nm rad$^{-1}$). We used kinematic and kinetic measures within inverse dynamics analysis to determine ankle, knee, and hip mechanics and surface electromyography (EMG) to characterize muscle activation of ankle plantar- and dorsi- flexors and knee flexors and extensors. We hypothesized that (i) incline walking with passive ankle exoskeletons would place a further requirement on the hip to produce positive power due to the reduced energy absorbed at the ankle, and therefore, reduced elastic energy stored within both the plantarflexors and exoskeleton device during incline walking, and (ii) decline walking with passive ankle exoskeletons would lead to an increased requirement of the knee to perform negative work when compared to no assistance.

## 2. Materials and Methods

### 2.1. Experimental Overview

Fifteen participants (eight male, seven female, $25 \pm 5$ years, $70 \pm 11$ kg, $172 \pm 8$ cm, mean $\pm$ sd) provided written informed consent in accordance with the Declaration of Helsinki before participating in this experiment (University of Queensland Human Ethics Approval #2019002595). All participants had no pre-existing neuromuscular disorder or recent history (<12 months) of lower-limb surgery or injury. Participants attended the laboratory for two sessions of experimentation. In the first session, participants completed a familiarization protocol. In this familiarization session, participants walked for five minutes at each exoskeleton stiffness (0, 50, 120, 220, and 280 Nm rad$^{-1}$) on level, incline (+5°), and decline (−5°) surfaces (Figure S1). The exoskeleton familiarization session was held no longer than 7 days before the second laboratory session.

During the second session, participants completed an exoskeleton-walking protocol. During this protocol, participants walked on an instrumented treadmill (FIT5, Bertec Inc., Columbus, OH, USA) at 1.25 m s$^{-1}$ on level, incline (+5°), and decline (−5°) grades with exoskeleton stiffness conditions 0, 50, 120, 220, and 280 Nm rad$^{-1}$ pseudo-randomized (Figure S1). We used 3D motion capture and force plates to measure lower-limb kinematics and kinetics, respectively, and surface EMG to record lower-limb muscle activations. All outcome measures were collected on the right leg.

### 2.2. Passive Ankle Exoskeleton Design

The exoskeleton used in this study consisted of custom-designed shank and shoe attachments linked via two aluminum bars, which rotated at the ankle joint (Figure 1a). Shank and shoe attachments were 3D printed in onyx—a micro carbon fiber-filled nylon material (Mark One, Markforged Inc., Waltham, MA, USA). Foam was attached in the areas of skin–device contact to avoid user discomfort. Exoskeleton clutch and device slack lengths (clutch onset angles) were set consistent with [5]. Clutches were machined as per the specification of [5,20], with some modifications to improve functionality. These include an enhanced pin-popper mechanism (by way of a ridge that connects both pin contact surfaces over the non-operational range of the ratchet) and bushings to enhance device durability (Figure 1b). This ratchet-paw system actuates based on the kinematics of the ankle joint, whereby the clutch engages at the end of the swing phase, immediately prior to heel contact, and releases at the end of the stance phase. The total mass of clutch was 58 g, and, assembled with the shoe attached (US size 10), the mass of one passive ankle exoskeleton was 500 g.

To alter device stiffness, steel coil spring elements were interchanged (Figure 1a). We used four springs of various stiffnesses (3.1, 7.2, 13.1, and 16.7 kN m$^{-1}$). Linear spring stiffnesses were determined from fixed-end spring extension experiments. The average moment arm of the ankle exoskeleton was 0.13 m, which resulted in average device rotational stiffnesses of approximately 50, 120, 220, and 280 Nm rad$^{-1}$. The range of device stiffnesses (0 (no spring), 50, 120, 220, and 280 Nm rad$^{-1}$) was selected to include stiffnesses within the optimal device stiffness range previously shown to reduce the metabolic cost of walking (at 1.25 m s$^{-1}$), as determined by [8] (50 Nm rad$^{-1}$) and [5] (180 Nm rad$^{-1}$). We note that there may have been differences in the effective stiffnesses between these devices owing to the overall design and attachment of the exoskeleton to the user.

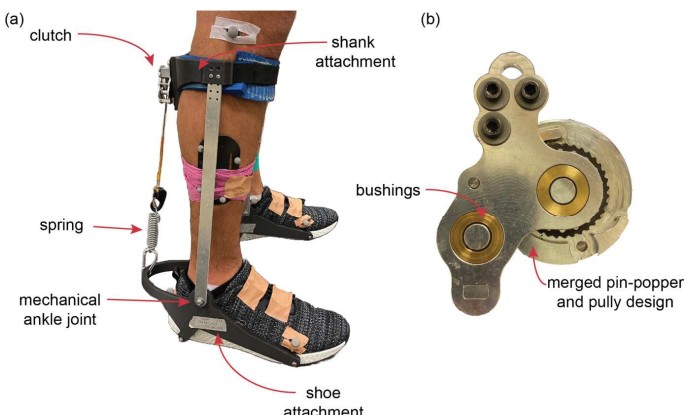

**Figure 1.** Exoskeleton frame and clutch design. (**a**) The exoskeleton used in this study consisted of custom-designed 3D-printed shank and shoe attachments that were linked via two aluminum bars, which rotated at the ankle joint. (**b**) Clutches were machined as per the specification of [5,19] with minor alterations, including an enhanced pin-popper mechanism (by way of a ridge that connects both pin contact surfaces over the non-operational range of the ratchet) and bushings to enhance device reliability. This ratchet-paw system actuates based on the kinematics of the ankle joint, whereby it is engaged at the end of swing phase, just before heel contact. With our minor alterations, the total mass of each clutch was 58 g.

## 2.3. Joint-Level Kinematics, Kinetics, and Mechanical Energetics

Three-dimensional lower-limb positions were recorded via an eight-camera motion capture system (120 Hz, Miqus, Qualisys AB, Göteborg, Sweden). Individual reflective markers were placed bilaterally on bony landmarks of the lower-limbs and pelvis. Custom 3D-printed rigid body clusters of four markers were secured to the left and right thigh and shank segments. This marker set was consistent with previous experimental protocols [21,22]. Participants were asked to ensure each foot was on a separate force plate of the instrumented treadmill during walking trials such that one 3D ground reaction force (GRF) vector could be attributed to each of the right and left lower-limbs (1200 Hz, FIT5, Bertec Inc., Waltham, MA, USA). Marker trajectories and force data were filtered with second-order low-pass Butterworth filters at 10 Hz.

OpenSim (v4.4) was employed to scale a musculoskeletal model based on a static standing calibration trial conducted for each participant [23,24]. Motion capture and force plate data from walking trials were utilized in inverse kinematics and dynamics analyses to determine the time-varying sagittal joint angles and moments at the ankle, knee, and hip. Experimental exoskeleton moments were estimated as the product of the experimentally determined exoskeleton spring stiffness and dynamic spring displacement during walking (measured using reflective marker displacement on the spring component of the device) and the average device moment arm (0.13 m). However, biological moments (owing to the action of muscles and tendons) and mechanical moments (owing to the action of exoskeletons) were not separated at the ankle joint during incline or decline walking trials due to increased noise in estimating dynamic spring displacements. All kinematic and kinetic time-varying group data were time normalized via interpolation to 100 points over the gait cycle and averaged over 10 strides. Joint and exoskeleton moments were normalized to each participant's body mass.

Instantaneous joint powers, which refer to the rate of work being performed by a joint, were determined as the product of joint angular velocities and joint moments. Positive joint power represents the action to extend the joint, and negative joint powers represent flexion. Joint powers were normalized to each participant's body mass. To determine joint work, the trapezium method was used to integrate joint power over periods of positive and negative work [2]. Negative and positive work at each individual joint were summed to determine the net joint work. Positive and negative work values were divided by stride time to calculate the total average positive and negative joint mechanical power, due to the

importance of capturing the power that is performed by the hip and knee during the swing phase [17]. Average net power was determined as the sum of positive and negative average power values at each joint. Positive, negative, and net joint work and power values were normalized to each participant's body mass.

*2.4. Muscle Activation*

Surface EMG was used to record muscle activity from SOL, medial gastrocnemius (GM), lateral gastrocnemius (GL), TA, vastus medialis (VM), vastus lateralis (VL), semitendinosus (ST), and biceps femoris (BF). The participant's skin was shaved, abraded, and cleaned with alcohol to reduce skin–electrode impedance. Muscle boundaries on the right leg were located using B-mode ultrasound (LV8-5L60N-2, Field of View 60 mm, Telemed UAB, Vilnius, LTU). Bipolar electrodes (2048 Hz, Trigno Avanti, inter-electrode distance: 10 mm, Delsys Inc., Natick, MA, USA) were placed over the muscle belly in the direction of muscle fascicles as per SENIAM guidelines. EMG signals were processed via a custom script (MATLAB 2022b, MathWorks Inc., Natick, MA, USA). First, signals were band-pass filtered using a second-order Butterworth filter (20–500 Hz). We visually checked the raw EMG data to identify movement artefacts or noise. The rectified EMG signals from 10 gait cycles were low-pass filtered at 12 Hz to determine an EMG envelope and then normalized to the maximum EMG activity for each muscle recorded during the no assistance level walking condition and averaged. We report average EMG over the whole stance phase of walking (0–60% gait cycle) and swing phase (60–100% gait cycle). Average EMG was calculated by dividing the integrated EMG (iEMG) by the time period over which it was integrated, where iEMG was determined as the time integral of the normalized EMG envelope averaged across each given period.

*2.5. Statistical Methods*

Linear mixed-effects (LME) models were used to determine the influence of exoskeleton stiffness (0, 50, 120, 220, and 280 Nm rad$^{-1}$) on joint kinematics, mechanical energetics (work and power), and muscle activation at each grade (0°, +5°, and −5°), separately. For all analyses, a within-participant design was used, including participant as a random factor using the lme.R function from the nlme package in R (R v4.2.2, R Foundation for Statistical Computing, Vienna, AUT) [25]. The glht.R function from the multcomp package was used to perform Tukey post hoc tests. Differences were considered significant at $p < 0.05$.

## 3. Results

*3.1. Level Walking*

The ankle was more plantarflexed during the stance phase of walking with increasing assistance ($p = 0.01$, Figure 2, Table S1, and Figure S2). Post hoc tests revealed a small, but significant, difference in ankle angle between the 0 and 220 Nm rad$^{-1}$ conditions (1.3 ± 0.4°, mean ± std. error). Peak positive biological ankle moments were reduced with exoskeleton assistance ($p < 0.001$, Figure 2). Post hoc analysis showed a difference in peak positive biological ankle moment between the 50 Nm rad$^{-1}$ (−0.09 ± 0.02 Nm kg$^{-1}$), 120 Nm rad$^{-1}$ (−0.07 ± 0.02 Nm kg$^{-1}$), 220 Nm rad$^{-1}$ (−0.1 ± 0.02 Nm kg$^{-1}$), and 280 Nm rad$^{-1}$ (−0.13 ± 0.02 Nm kg$^{-1}$) conditions, compared to the no stiffness condition (0 Nm rad$^{-1}$). However, peak positive biological power differed only between the 280 Nm rad$^{-1}$ condition, compared to no assistance ($p = 0.004$, Figure 2), with a difference in peak positive biological power of 0.6 ± 0.2 W kg$^{-1}$. Increasing exoskeleton stiffness led to a greater range of motion at the knee during the stance phase of walking ($p = 0.014$, Figure 2 and Table S1). Post hoc analysis showed a small, but significant, difference in average knee range of motion between the 50 Nm rad$^{-1}$ (2.1 ± 0.6°), 120 Nm rad$^{-1}$ (1.6 ± 0.6°), and 220 Nm rad$^{-1}$ (1.3 ± 0.4°) conditions, compared to the no stiffness condition (0 Nm rad$^{-1}$). At the hip, increasing exoskeleton stiffness resulted in greater peak positive hip power ($p = 0.015$, Figure 2), with post hoc tests showing a difference between the 0 and 50 Nm rad$^{-1}$

$(0.19 \pm 0.05$ W kg$^{-1}$) conditions only. There was no influence of exoskeleton stiffness on average ankle, knee, or hip positive, negative, or net power during level walking (Figure 3).

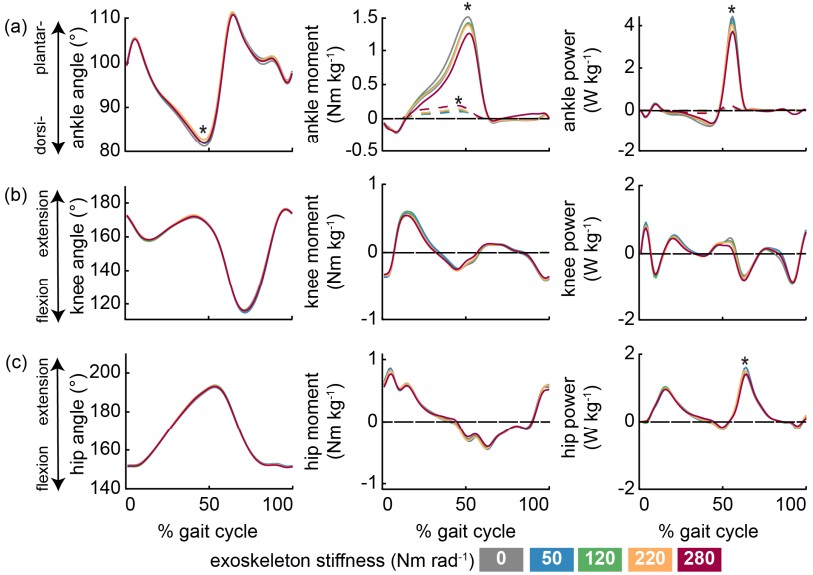

**Figure 2.** Passive ankle exoskeletons influenced ankle and knee kinematics and ankle kinetics during level walking. Group mean ankle (**a**), knee (**b**), and hip (**c**) joint angles (°) (**left**), moments (Nm kg$^{-1}$) (**center**), and powers (W kg$^{-1}$) (**right**). Group mean curves are time normalized to the 0-100% of the gait cycle. Exoskeleton stiffness conditions (0, 50, 120, 220, and 280 Nm rad$^{-1}$) are denoted by color. Estimated exoskeleton moments and powers are shown with dashed lines in the corresponding exoskeleton condition color on the ankle moment ((**a**) (**center**)) and ankle power ((**a**) (**right**)) plots. The black dashed line indicates zero moment or power. A main effect of exoskeleton assistance on peak joint angle, moment, or power are denoted by * ($p < 0.05$).

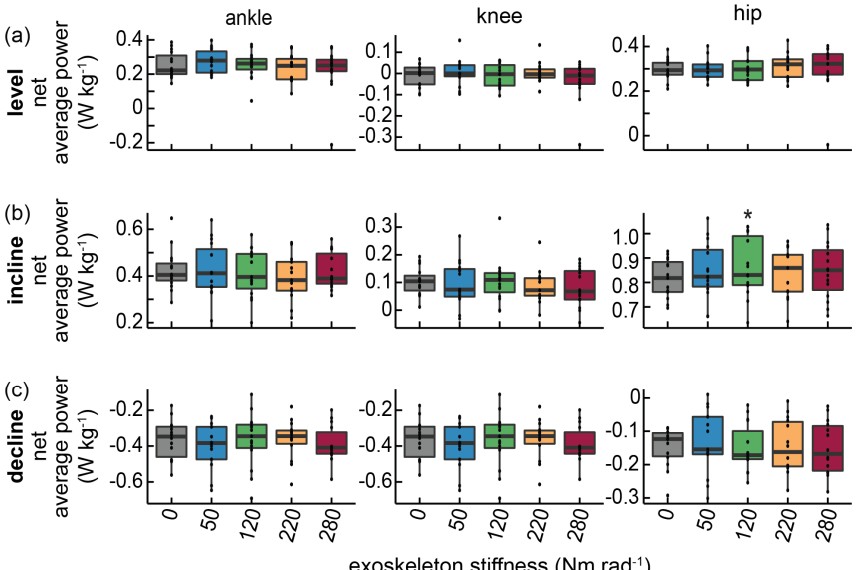

**Figure 3.** Passive ankle exoskeletons influence average net knee and hip joint power during incline, but not level or decline walking. Ankle (**left**), knee (**middle**), and hip (**right**) average power during (**a**) level, (**b**) incline, and (**c**) decline walking. Box and whisker plots show the median, with hinges representing the first and third quartiles; whiskers represent 1.5× the inter-quartile range. Exoskeleton stiffness conditions (0, 50, 120, 220, and 280 Nm rad$^{-1}$) are denoted by color. Significant differences from the 0 Nm rad$^{-1}$ condition according to the Tukey post hoc are denoted by * ($p < 0.05$).

During level walking, increasing exoskeleton stiffness led to a reduction in average SOL activation during the stance phase of walking ($p = 0.022$, Figures 4 and 5). Post hoc analysis showed that when compared to the no stiffness condition (0 Nm rad$^{-1}$), exoskeletons reduced SOL activation by $11.0 \pm 3.7\%$ at the 50 Nm rad$^{-1}$ condition, $12.9 \pm 3.9\%$ at the 120 Nm rad$^{-1}$ condition, and $11.2 \pm 3.8\%$ at the 220 Nm rad$^{-1}$ condition. In addition, increasing device stiffness led to an increase in average TA activation during the swing phase of walking ($p = 0.012$, Figures 4 and 5). Post hoc analysis showed that when compared to the no stiffness condition (0 Nm rad$^{-1}$), exoskeletons increased TA activation by $16.7 \pm 5.4\%$ at the 50 Nm rad$^{-1}$ condition, $16.2 \pm 5.4\%$ at the 120 Nm rad$^{-1}$ condition, and $18.6 \pm 5.5\%$ at the 220 Nm rad$^{-1}$ condition.

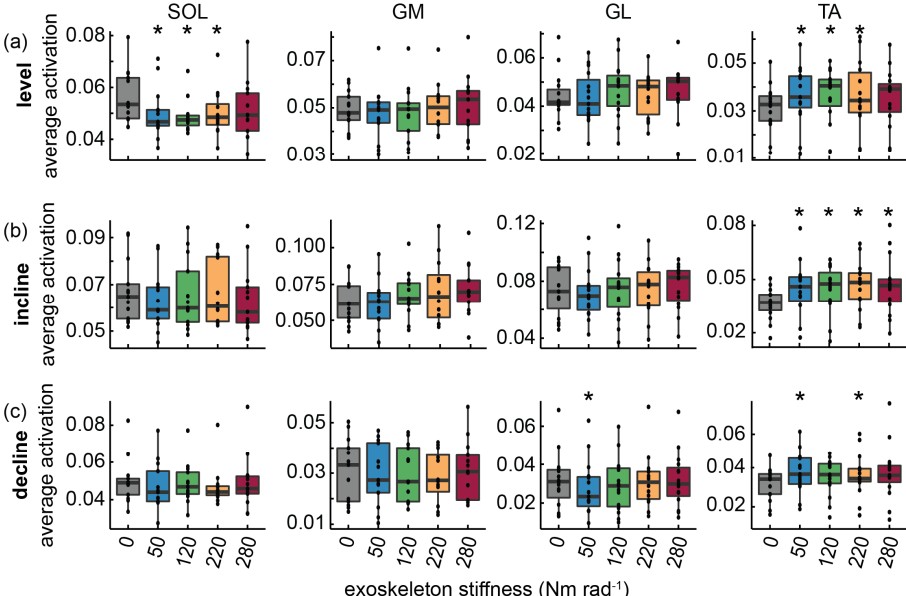

**Figure 4.** Passive ankle exoskeletons influence average muscle activation in a grade-dependent manner. Soleus (SOL), medial gastrocnemius (GM), and lateral gastrocnemius (GL) average muscle activation during stance, and tibialis anterior (TA) average muscle activation during swing of (**a**) level, (**b**) incline, and (**c**) decline walking. Box and whisker plots show the median, with hinges representing the first and third quartiles; whiskers represent 1.5× the inter-quartile range. Exoskeleton stiffness conditions (0, 50, 120, 220, and 280 Nm rad$^{-1}$) are denoted by color. Significant differences from the 0 Nm rad$^{-1}$ condition according to the Tukey post hoc are denoted by * ($p < 0.05$).

### 3.2. Incline Walking

Participants walked in more plantarflexed postures during the stance phase of incline walking with increasing device stiffness ($p = 0.003$, Table S1 and Figure S3). Post hoc tests uncovered increased plantarflexion compared to the no stiffness condition (0 Nm rad$^{-1}$) at the 120 Nm rad$^{-1}$ ($1.2 \pm 0.4°$), 220 Nm rad$^{-1}$ ($1.6 \pm 0.4°$), and 280 Nm rad$^{-1}$ ($1.3 \pm 0.4°$) conditions, respectively. Increasing exoskeleton stiffness was associated with reduced range of motion at the knee during stance ($p = 0.024$, Table S1 and Figure S3). Post hoc analysis showed this reduction in joint range of motion between the 50 Nm rad$^{-1}$ ($2.1 \pm 0.7°$) and 280 Nm rad$^{-1}$ ($2.3 \pm 0.7°$) conditions, compared to the no stiffness condition (0 Nm rad$^{-1}$). More negative work was performed at the knee with increasing device stiffness ($p = 0.019$, Table S3) such that at the 50 and 220 Nm rad$^{-1}$ conditions, the knee performed $14.5 \pm 4.6\%$ and $15.2 \pm 4.6\%$ more negative work, compared to no assistance. In turn, this led to a concurrent increase in average negative power at the knee ($p = 0.03$), where at the 50 and 220 Nm rad$^{-1}$ conditions, the knee performed $14.7 \pm 4.9\%$ and $15.6 \pm 4.9\%$ more average negative power, compared to no assistance. At the hip, assistance led to greater peak positive hip moments ($p = 0.046$), with post hoc tests revealing a significant difference between 0 and 120 Nm rad$^{-1}$ ($0.09 \pm 0.28$ Nm kg$^{-1}$). More positive work was

performed at the hip with increasing assistance ($p$ = 0.034), with post hoc analysis revealing a $6.1 \pm 1.8\%$ increase at the 120 Nm rad$^{-1}$ condition, compared to no assistance. The increase in positive work led to a $6.1 \pm 1.8\%$ increase in net hip work at the 120 Nm rad$^{-1}$ condition compared to no assistance (Figure 3). Compared to the no stiffness condition, peak positive power, average positive hip power, and average net hip power were $11.1 \pm 3.4\%$, $5.8 \pm 1.9\%$. and $5.8 \pm 1.9\%$ greater at the 120 Nm rad$^{-1}$ condition, respectively (all: $p \leq 0.027$).

Increasing exoskeleton stiffness led to an increase in average TA activation during the swing phase of incline walking ($p \leq 0.025$, Figures 4 and 5). TA activation increased by $24.9 \pm 5.3\%$, $23.8 \pm 5.3\%$, $29.3 \pm 5.4\%$, and $29.1 \pm 5.3\%$ at the 50, 120, 220, and 280 Nm rad$^{-1}$ conditions, compared to no assistance, respectively.

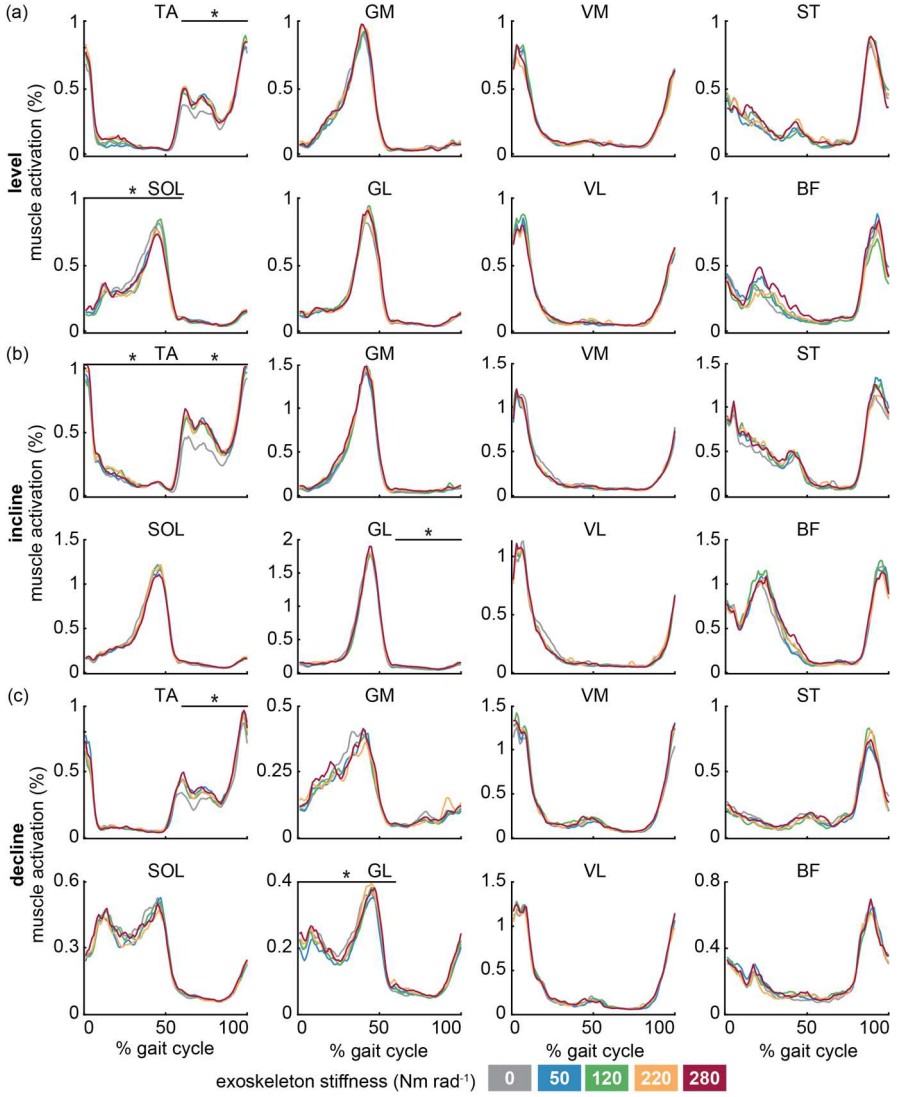

**Figure 5.** The grade dependent influence of passive ankle exoskeletons on lower-limb neuromuscular control during level, incline, and decline walking. Time-varying group mean muscle activation for tibialis anterior (TA), medial gastrocnemius (GM), lateral gastrocnemius (GL), soleus (SOL), vastus lateralis (VL), vastus medialis (VM), biceps femoris (BF), and semitendinosus (ST) during (**a**) level, (**b**) incline, and (**c**) decline walking. Group mean curves are time normalized to 0–100% of the gait cycle. Exoskeleton stiffness conditions (0, 50, 120, 220, and 280 Nm rad$^{-1}$) are denoted by color. * denotes a main effect of exoskeleton assistance ($p < 0.05$) on average muscle activation during the stance (0-60%) and swing (60-100%) phases.

*3.3. Decline Walking*

Additional stiffness at the ankle did not alter lower-limb kinematics or moments during decline walking. Yet, exoskeleton stiffness led to a reduction in peak positive ankle power ($p$ = 0.022, Figure S4). Post hoc tests showed a significant reduction in peak ankle power at the 280 (0.38 $\pm$ 0.12 W kg$^{-1}$) Nm rad$^{-1}$ condition, compared to no stiffness. There was no influence of assistance on negative, positive or net, work or average power at any lower-limb joint (Table S3).

Increasing exoskeleton stiffness led to a reduction in GL activation during the stance phase of decline walking ($p$ = 0.016, Figures 4 and 5), whereby activation decreased by 10.9 $\pm$ 3.8% at the 50 Nm rad$^{-1}$ condition, compared to the no stiffness condition. TA average activation during the swing phase increased with exoskeleton stiffness ($p$ = 0.01, Figures 4 and 5). Post hoc tests revealed a significant increase in TA activation of 21.3 $\pm$ 6.1% and 20.5 $\pm$ 6.1% at the 50 and 220 Nm rad$^{-1}$ conditions compared to the no stiffness condition, respectively.

## 4. Discussion

This study explored how passive ankle exoskeletons influence lower-limb mechanical energetics and neuromuscular control during level, incline, and decline walking. Our results demonstrated that the influence of passive ankle exoskeleton on the lower-limb neuromechanics of walking varies with grade. In support of hypothesis (i), walking with passive ankle exoskeletons placed a further requirement on the hip to produce positive power. However, hypothesis (ii) was not supported by our results, as exoskeleton assistance during decline walking had no influence on knee mechanical energetics (Figure 3), but rather participants responded via alterations in TA activation and reductions in peak ankle power.

The neuromechanical changes observed during level walking with ankle exoskeleton assistance were largely consistent with previous studies [5,8]. For example, increased device stiffness was associated with increases in ankle plantarflexion and a reduction in SOL muscle activation during the stance phase, as well as increases in TA activation during the swing phase of walking [5,8]. Further, average SOL activation over the stance phase displayed a 'bowl-shaped' relationship with device stiffness, whereby a maximum ~13% reduction in SOL activation occurred at 120 Nm rad$^{-1}$ (Figure 4a). The 'bowl-shaped' relationship between SOL activation and device stiffness is consistent with the relationship between the net metabolic rate and stiffness previously demonstrated, whereby stiffnesses between 50 and 80 Nm rad$^{-1}$ minimized the metabolic rate of walking [5,8]. Here, we find that SOL activation was minimized at a slightly higher intermediate stiffness, although this difference is likely due to additional compliance in our device owing to 3D-printed, rather than carbon fiber, materials.

As suggested by others [17], incline walking with passive ankle exoskeletons likely results in marginal mechanical benefit for the user. Spring-clutch ankle exoskeletons harvest energy from the ankle, and then return that energy at push off. During incline walking, the hip is the dominant site of positive power generation [17]. Muscles that cross the hip have architectures that are well suited to performing positive work on the COM, compared to muscles that cross the ankle. In addition, the capacity of springy ankle exoskeletons to store energy at the ankle is reduced when walking uphill because negative ankle power is reduced [17]. In support of hypothesis (i), our results demonstrate that walking uphill with ankle exoskeletons leads to an increase in positive work and average power at the hip. These alternations in proximal joint function may be linked to the increased energy absorption at the knee when walking uphill with ankle exoskeletons—albeit the lack of systematic differences with increasing device stiffness suggests there are likely large interaction effects. The increase in negative work and average negative power observed at the knee may be the result of complex inter-joint interaction with the ankle, whereby the negative work functions to load the exoskeletons' springy components during incline walking [26]. An integral part of this complex interaction is clutch actuation. In this study, clutch engagement was set for the level walking condition and not reset for incline

or decline walking conditions. During incline walking, clutch actuation likely occurred during the swing phase, as evidenced by a trend towards more plantarflexed postures from the mid-swing phase of walking. Unintentionally, by starting to harvest energy during the swing phase, the capacity of the device to assist incline walking may be increased in comparison to a clutch that began to store energy during stance.

Passive ankle exoskeletons did not lead to reductions in activation of any lower-limb muscle during incline walking. A closer inspection of the time-varying SOL activation profiles when walking uphill with exoskeleton assistance demonstrated average SOL activation during mid-stance (20-40% of stance phase), followed a 'bowl-shaped' relationship with device stiffness, whereby SOL activation reduced by $10.5 \pm 3.8\%$ at the 120 Nm rad$^{-1}$ condition compared to no assistance ($p = 0.039$). In addition, average activation of the knee extensors VM and VL at mid-stance was reduced by up to $23.7 \pm 7.5\%$ and $27.1 \pm 8.1\%$, respectively, at the 50 Nm rad$^{-1}$ condition (both: $p \leq 0.026$) compared to no assistance. Perhaps the reduction in VM and VL activation is linked to the observed reductions in knee range of motion. However, these neuromuscular benefits owing to assistance are likely negated by increases in TA activation during the swing phase of walking. TA average activation increased by ~25-29% with increasing device stiffness ($p < 0.001$). The activity of muscles that cross the hip also likely increased, but EMG data was not captured in this study for these muscles. Similar muscle activation results for VL [4] and TA [4,27] have been observed during uphill walking with powered ankle exoskeletons. However, Galle et al. [4] showed that when walking uphill with a powered ankle exoskeleton, users increased total ankle work to reduce work at the hip, which does not appear to be the case when walking uphill in passive devices.

Walking down sloped surfaces with ankle exoskeleton assistance did not lead to increases in energy absorption at the knee, in disagreement with our hypothesis (ii). Passive ankle exoskeletons require appropriate clutch actuation timing to enable effective energy storage and return via their elastic elements. Given that clutch engagement was not reset for sloped conditions, during decline walking, clutch actuation may have occurred towards late-stance (or not at all). Thus, the additional energy of the exoskeleton (stored in the spring component) was limited and could effectively be managed by the ankle by increasing TA activity (Figures 4 and 5) and reducing peak positive ankle power.

The results from this work suggest that walking on sloped surfaces with exoskeleton assistance may require alterations to device designs, which could potentially be accommodated via two strategies. First, an active–passive device that can modify slack length (*i.e.,* the timing of actuation) based on environmental conditions and prior gait events. Such a design may avoid the unwanted increases in lower-limb muscle activation during incline and decline walking with assistance. Second, the results from our investigation suggest that the optimum ankle exoskeleton stiffness is likely dependent on the slope of the walking surface. The requirements placed on the lower-limb to load springy exoskeletons could be reduced by using a variable stiffness device, whereby the device could shift to a low-stiffness mode during incline walking and a higher-stiffness mode when walking on level ground.

We would like to acknowledge that device compliance limited our ability to measure exoskeleton moments during some walking conditions and likely reduced the effective stiffness of our device. Incline and decline walking with assistance caused deformation in other components of the device, which limited our ability to estimate device moments from measures of spring extension. As such, we report exoskeleton moments during level (Figure 2), but not incline or decline walking because of the challenges associated with accounting for this additional device compliance when determining moments from spring displacements. Partitioning the mechanical contributions to ankle moment, work, and power yields important insights into the user–device interaction. Future studies could incorporate a force sensor in series with the springy components of our exoskeleton device [5,28]. However, considerations owing to the added mass and potential for the force sensor to disrupt surface EMG signals are necessary. In addition, this experimental design

did not include metabolic data. The metabolic benefit of this clutch design in physical [5] and emulator systems [8] has been well established in level walking. Although metabolic data would been useful in interpreting if there is indeed an energetic cost of using the device under the incline and decline conditions, this outcome measure was not a focus of this work.

## 5. Conclusions

We have provided important insights into how passive ankle exoskeletons influenced the mechanical energetics and neuromuscular control of the lower-limb during walking on sloped surfaces. These results demonstrate how passive ankle exoskeletons require a shift in mechanical energetics across the knee and hip to both push the COM uphill and effectively load the springy exoskeleton. However, it should be noted that alterations in lower-limb mechanics and muscle activation were not always more pronounced with increasing exoskeleton stiffness at any walking conditions, and there were no apparent linear trends (increase or decrease) in these outcome measures. This highlights that the interaction between assistive devices and biological tissues is complex, with factors spanning device design, task performed, user, and environment [29]. Indeed, future work is required to probe these interactions for the most effective design of devices for different environments and users. Further investigation into the influence of wearable assistive technologies in the more real-world locomotor conditions of daily life could extend this experimental paradigm to characterize the interaction between walking speed and grade or the influence of individual variation in muscle–tendon properties on the response to assistance.

**Supplementary Materials:** The following supporting information can be downloaded at: https://www.mdpi.com/article/10.3390/machines11121071/s1, Table S1: Lower-limb kinematics during level, incline, and decline walking, with and without exoskeleton assistance; Table S2: Lower-limb kinetics during level, incline, and decline walking, with and without exoskeleton assistance; Table S3: Lower-limb positive and negative work during level, incline, and decline walking, with and without exoskeleton assistance; Figure S1: An illustration of the experimental paradigm; Figure S2: The influence of passive ankle exoskeletons on ankle, knee, and hip joint mechanics during level walking; Figure S3: The influence of passive ankle exoskeletons on ankle, knee, and hip joint mechanics during incline walking; Figure S4: The influence of passive ankle exoskeletons on ankle, knee, and hip joint mechanics during decline walking.

**Author Contributions:** J.L.W., G.A.L., and T.J.M.D. conceived the study and designed the experimental protocol; J.L.W. carried out the experiments; J.L.W. analyzed the data and drafted the manuscript; T.J.M.D. and G.A.L. edited the manuscript. All authors have read and agreed to the published version of the manuscript.

**Funding:** J.L.W. is supported by Australian Government RTP stipend and RTP Tuition Fee Offset Scholarships. This work was also supported by a Queensland Defence Science Alliance, HDR Capability Demonstration grant to J.L.W. The funders had no role in the study design, data collection and analysis, decision to publish, or preparation of the manuscript.

**Data Availability Statement:** Participant average kinematic, kinetic and muscle activation data ($n = 15$) for each grade ($0°$, $+5°$, $-5°$) and exoskeleton stiffness (0–280 Nm rad$^{-1}$) are available upon request.

**Acknowledgments:** We acknowledge Matt Richards for assistance with data collection. We thank Gregory Sawicki, for support in clutch design and implementation, and Sam Grieve at The University of Queensland Innovate Makerspace, for assistance in clutch manufacturing.

**Conflicts of Interest:** The authors declare no conflict of interest.

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
