# Peer review of "Exploring the Impact of Passive Ankle Exoskeletons on Lower-Limb Neuromechanics during Walking on Sloped Surfaces: Implications for Device Design"

_machines, doi:10.3390/machines11121071_

Round 1

Reviewer 1 Report

Comments and Suggestions for Authors

The paper presents an interesting topic, evaluating the performance of a custom designed ankle exoskeleton. It uses multimodal methods, including motion capture systems and EMG sensing on one leg to assess the exoskeleton. The paper then gives suggestions for future implementations of exoskeletons.

The introduction is easy to follow, and - as the rest of the paper - well written. However, I suggest the authors to back their literature review in the introduction with more references. This would also make a stronger case of justifying the choices made further on for the experimental set up and methods used to analyse the collected data.

The experimental set up is interesting, although I would have been keen to see more parameters tested, more comparisons carried out. For example, why are the additional sensing methods only used on the right leg? Why the right leg? I would also have more questions on the participant demographics. How, for example, does their weight affect the outcome of the study? The presented data shows little variation in weight (though that may be relative to height), and it would be interesting to discuss the range more, including more “extreme” participants (e.g. very tall and thin, or very short and big, etc). 

It is not clear to me, whether the main contribution is the design of the exoskeleton, in which case the paper could be framed in a slightly different way and the technical details should be elaborated more; or the evaluation of an exoskeleton that has already been presented to the scientific community.

Concluding with design implementations is a good and useful contribution. However, I would describe them more as guidelines or a checklist so that other researchers in the field can follow them. In a related context, when making suggestions of how in detail the design of the exoskeleton can be improved, make it part of the discussion rather than the conclusion. The conclusion should not contain too much repetition from that.

Some other, minor comments:

- Figure 2-5 could be smaller, as clear enough. 

- instead, there could be an additional figure illustrating the experimental set up more, as well as one showing details of the exoskeleton on the crucial areas.

In conclusion, I suggest the above considerations to be implemented and the paper to be revised before publication. 

Reviewer 2 Report

Comments and Suggestions for Authors

Introduction:

The introduction provides a comprehensive overview of the existing literature and provides a broad understanding of the subject matter. The introduction emphasizes relevance and potential implications of the research findings.

Methods:

Please elaborate on how the stiffness of the device was altered. It is unclear from the existing text how the stiffness was changed.

Results

I recommend shortening the results section slightly by editing it to present only the key findings. Try to streamline the presentation of the results and move secondary details to the appendix or supplementary materials. Focusing on the main results will help highlight the most important outcomes and strengthen the overall clarity of this section.

Reviewer 3 Report

Comments and Suggestions for Authors

This is an article which evaluates the impact of passive ankle exoskeletons on lower limbs neuromechanics during walking on surfaces with different slopes.

The text is well written, and the English is good. Moreover, the study has some merits attributable above all to the research question at the base of the study and the to the rigor with which they followed the methodology.

On the other hand, the study needs some revisions, to make the Results, Discussion and Conclusions more connected to the obtained results.

It is also suggested to add a section with the limitations of the study and possible future developments to overcome them.

Attached is the revised PDF of the article with detailed comments on the requested small revisions.

Round 2

Reviewer 3 Report

Comments and Suggestions for Authors

The article can now be accepted in the reviewed form.

Thanks to the authors.